# The Chengjiang Biota inhabited a deltaic environment

Farid Saleh [1,2,9 ✉], Changshi Qi[1,2,3,9], Luis A. Buatois[4], M. Gabriela Mángano[4], Maximiliano Paz[4], Romain Vaucher [5,6], Quanfeng Zheng[3], Xian-Guang Hou[1,2], Sarah E. Gabbott[7] & Xiaoya Ma [1,2,8 ✉]

The Chengjiang Biota is the earliest Phanerozoic soft-bodied fossil assemblage offering the most complete snapshot of Earth's initial diversification, the Cambrian Explosion. Although palaeobiologic aspects of this biota are well understood, the precise sedimentary environment inhabited by this biota remains debated. Herein, we examine a non-weathered core from the Yu'anshan Formation including the interval preserving the Chengjiang Biota. Our data indicate that the succession was deposited as part of a delta influenced by storm floods (i.e., produced by upstream river floods resulting from ocean storms). Most Chengjiang animals lived in an oxygen and nutrient-rich delta front environment in which unstable salinity and high sedimentation rates were the main stressors. This unexpected finding allows for sophisticated ecological comparisons with other Burgess Shale-type deposits and emphasizes that the long-held view of Burgess Shale-type faunas as snapshots of stable distal shelf and slope communities needs to be revised based on recent sedimentologic advances.

[1] Yunnan Key Laboratory for Palaeobiology, Institute of Palaeontology, Yunnan University, Kunming, China. [2] MEC International Joint Laboratory for Palaeobiology and Palaeoenvironment, Institute of Palaeontology, Yunnan University, Kunming, China. [3] State Key Laboratory of Palaeobiology and Stratigraphy, Nanjing Institute of Geology and Palaeontology and Center for Excellence in Life and Paleoenvironment, Chinese Academy of Sciences, Nanjing, Jiangsu, China. [4] Department of Geological Sciences, University of Saskatchewan, Saskatoon, SK, Canada. [5] Applied Research in Ichnology and Sedimentology (ARISE) Group, Department of Earth Sciences, Simon Fraser University, Burnaby, BC, Canada. [6] Institute of Earth Sciences (ISTE), University of Lausanne, Geopolis, Lausanne, Switzerland. [7] School of Geography, Geology and Environment, University of Leicester, Leicester, LE, UK. [8] Centre for Ecology and Conservation, University of Exeter, Penryn, UK. [9] These authors contributed equally: Farid Saleh and Changshi Qi. ✉email: farid.nassim.saleh@gmail.com; x.ma2@exeter.ac.uk

The Chengjiang Biota (Cambrian Stage 3, China) is an exceptionally preserved fossil assemblage within which many taxa reveal exquisite details of both external and internal anatomical features[1–3]. It is also the oldest site known for preserving an extremely diverse assemblage of eumetazoans[4–6]. For instance, the number of taxa recorded in the Chengjiang Biota is 300% higher than in the Sirius Passet, which is of approximatively the same age as the Chengjiang Biota (Cambrian Stage 3, Greenland)[7]; and 27% higher than the most famous site with exceptional fossil preservation, the Burgess Shale Walcott Quarry (Wuliuan, Canada)[8]. Therefore, the Chengjiang Biota is important for our understanding of the Cambrian Explosion and its ecological complexity[1–3].

Despite hundreds of research articles describing taxa from the Chengjiang Biota[9–12] and their preservation[13–20], there is no consensus interpretation on the palaeoenvironment in which animals lived, died, and were preserved. A variety of depositional environments have been invoked ranging from tidally influenced estuaries and embayments to wave-dominated shoreface-offshore complexes and shelf settings affected by turbidity currents[21–28]. One of the factors that may account for these diverse interpretations is that, to date, environmental interpretations have been based on evidence from weathered outcrops[21–28], which suffer from information loss of the fine-scale sedimentary structures that are essential for inferring diagnostic characteristics of flow dynamics. A lack of understanding of palaeoenvironmental contexts presents a significant problem for our interpretation of the composition and ecology of animal life in the Chengjiang Biota and for comparisons of the Chengjiang Biota with other Cambrian Lagerstätten, such as the Emu Bay Shale, the Sirius Passet, and the Burgess Shale. Establishing the palaeoenvironmental context for Lagerstätten is critical as it allows an appreciation of the type of setting in which animals were evolving, the environmental constraints on communities, and ecology and insights into how preservation may have operated.

The investigation of the sedimentology of fine-grained deposits, similar to the Chengjiang Biota, has experienced a paradigm shift during the last couple of decades[29–34]. Many mudstone-dominated successions originally regarded as having formed owing to slow suspension fallout are now reinterpreted as having been produced by a complex array of depositional mechanisms, such as hyperpycnal flows, fluid muds, ocean floods, turbidity currents, and bottom currents, among other processes[35–41]. This recent way of investigating and interpreting fine-grained systems has rarely been applied systematically to the study of Burgess Shale-type deposits preserving soft anatomies in the Cambrian.

Here, we apply the latest advances in knowledge gained from sediment flow experiments, depositional modeling, and field and subsurface data to the analysis of fresh, non-weathered core material to describe the diagnostic characteristics of the Chengjiang Biota deposit and assess its sedimentary environment. This study shows that the interval hosting the Chengjiang Biota, one of the archetypal Burgess Shale-type deposits, was formed by a variety of flow types that indicate the influence of a mixed river- and wave-influenced delta. This establishes the palaeoenvironmental context of the Chengjiang Biota and provides insights into the environmental tolerance of Burgess Shale-type animals.

## Results and discussion

**Sedimentary flow deposits.** The core intersects all four members of the Yu'anshan Formation, from top to bottom: the Upper Siltstone Member, the Maotianshan Shale Member, the Black Carbonaceous Member, and the Black Siltstone Member (Fig. 1). For the description of the fine-grained deposits in the core we use the classification of Lazar et al.[34]. The following types of deposits have been recognized.

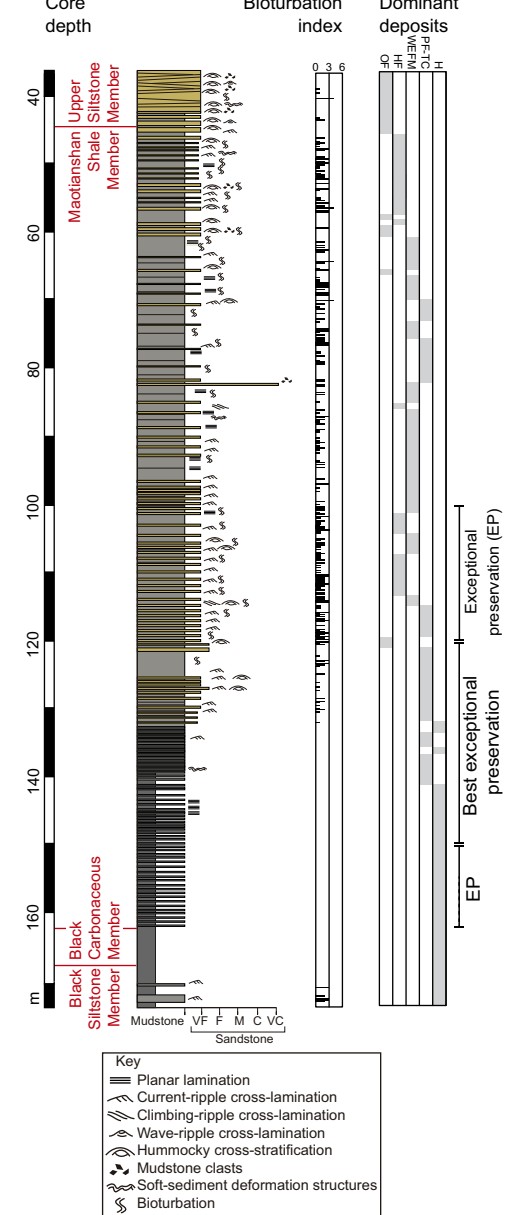

**Fig. 1 Stratigraphic succession of the Yu'anshan Formation, showing lithologic log, bioturbation index, and the distribution of the dominant deposits.** *OF* oscillatory-flow deposits, *HF* hyperpycnal flow deposits, *WEFM* wave-enhanced fluid mud deposits, *PF-TC* plug flow, low-density turbidity current deposits, and *H* hemipelagic deposits. *VF* very fine, *F* fine, *M* medium, *C* coarse, *VC* very coarse. Note the clear large-scale coarsening- and thickening-upward trend that is common in wave-influenced shallow-marine depositional systems, including most notable deltas. The so-called "Upper Siltstone Member" is actually sandstone-dominated.

Oscillatory-flow deposits: These deposits consist of medium-to-thick-bedded, erosionally based, parallel-laminated, and hummocky cross-stratified, well-sorted, very fine- to fine-grained sandstone (Fig. 2a, Supplementary Fig. 1a–d). Hummocky cross-stratification is of isotropic type and amalgamated. Discrete layers may show normal grading. These beds may pass upwards into intervals characterized by combined flow and oscillatory ripple cross-lamination. Very thin mudstone interbeds are present locally (Fig. 1, Supplementary Fig. 1a, d). Scour surfaces and gutter casts, although hard to identify in the core, have been observed locally being typically filled with low-angle to

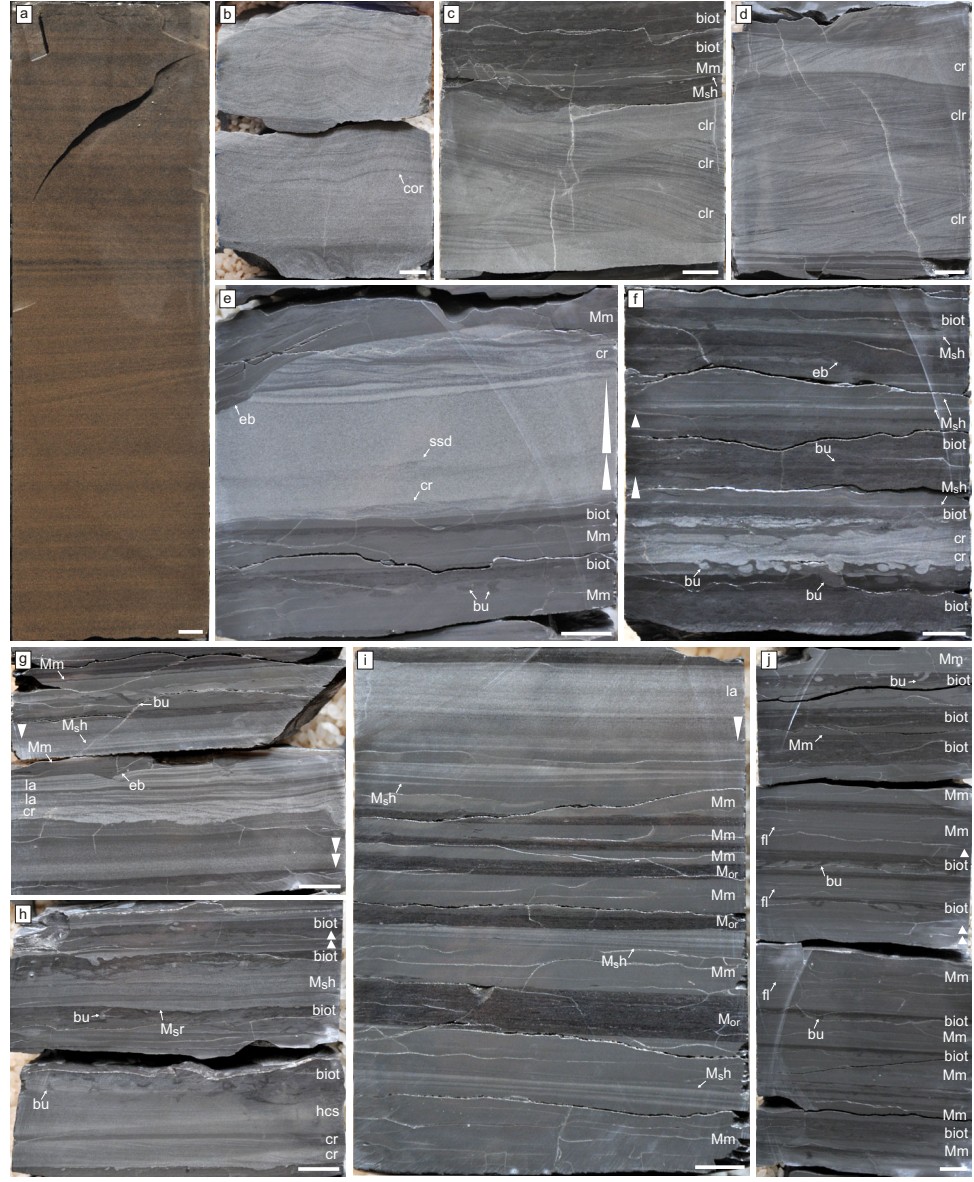

**Fig. 2 Core photographs of the different deposits observed in the Yu'anshan Formation. All scale bars represent 10 mm. a**, **b** Hummocky cross-stratified and climbing oscillation-ripple cross-laminated (cor), fine-grained sandstone of the oscillatory-flow deposits. Note strongly aggradational component of oscillatory climbing ripples in **b**. **c**, **d** Current-ripple (cr) and climbing-ripple (clr) cross-laminated, very fine-grained sandstone, interbedded with bioturbated intervals (biot), representing hyperpycnal or turbidity flow deposits. Massive fine mudstone (Mm) and coarse mudstone laminae (M$_s$h) are interbedded. **e** Normal-graded (white triangles) and current-ripple cross-laminated (cr), fine-grained sandstone with minor soft-sediment deformation structures (ssd) of the hyperpycnal flow deposits, interbedded with massive mudstone (Mm) with erosive bases (eb) typical of the wave-enhanced fluid mud deposits. Burrows (bu) occur on top of the mudstone beds. **f** Rippled fine-grained sandstone (cr) with sharp-lined firmground burrows (bu) penetrating from the base of a hyperpycnal flow layer into an intensely bioturbated mudstone initially colonized under softground conditions. Fluid mud deposits with normal-graded mudstone (white triangles), coarse mudstone laminae (M$_s$h), and erosive bases (eb) are observed, interbedded with bioturbated intervals (biot). **g–i** Low-angle cross-laminated (la), current-ripple cross-laminated (cr), and hummocky cross-stratified (hcs), very fine-grained sandstone of the hyperpycnal flow deposits, interbedded with fluid mud intervals with erosive bases (eb). These beds alternate with successions of plug flow and low-density turbidity current deposits consisting of massive mudstone (Mm) with some parallel coarse mudstone laminae (M$_s$h), current-ripple cross-laminated coarse mudstone (M$_s$r), and normal and inverse grading (white triangles), and with dark, organic-rich mudstone (M$_{or}$) of the hemipelagic deposits. Note bioturbation structures (bu) and bioturbated intervals (biot) at the top of beds. **j** Massive mudstone (Mm) of the plug flow and low-density turbidity current deposits, with local normal grading (white triangles) and faint lamination (fl), locally intercalated with bioturbated intervals (biot), the latter representing the hemipelagic deposits. Burrows (bu) can be observed towards the top of mudstone beds.

hummocky cross-stratified sandstone (Supplementary Fig. 1d). Soft-sediment deformation structures, such as convolute lamination and ball-and-pillow, are present (Supplementary Fig. 1a, c). Bioturbation is generally absent (Bioturbation Index: BI 0), although thin (<3 cm) mottled intervals characterized by shallow-tier undetermined trace fossils (BI = 3) occur locally

(Supplementary Fig. 1b). These deposits have only been recorded in the Upper Siltstone Member (Fig. 1). Parallel lamination is interpreted as corresponding to the upper-flow regime, whereas hummocky cross-stratification is inferred to have been produced by intense oscillatory flows during storms[42–44]. Combined flow ripple cross-lamination indicates the interplay of oscillatory flows

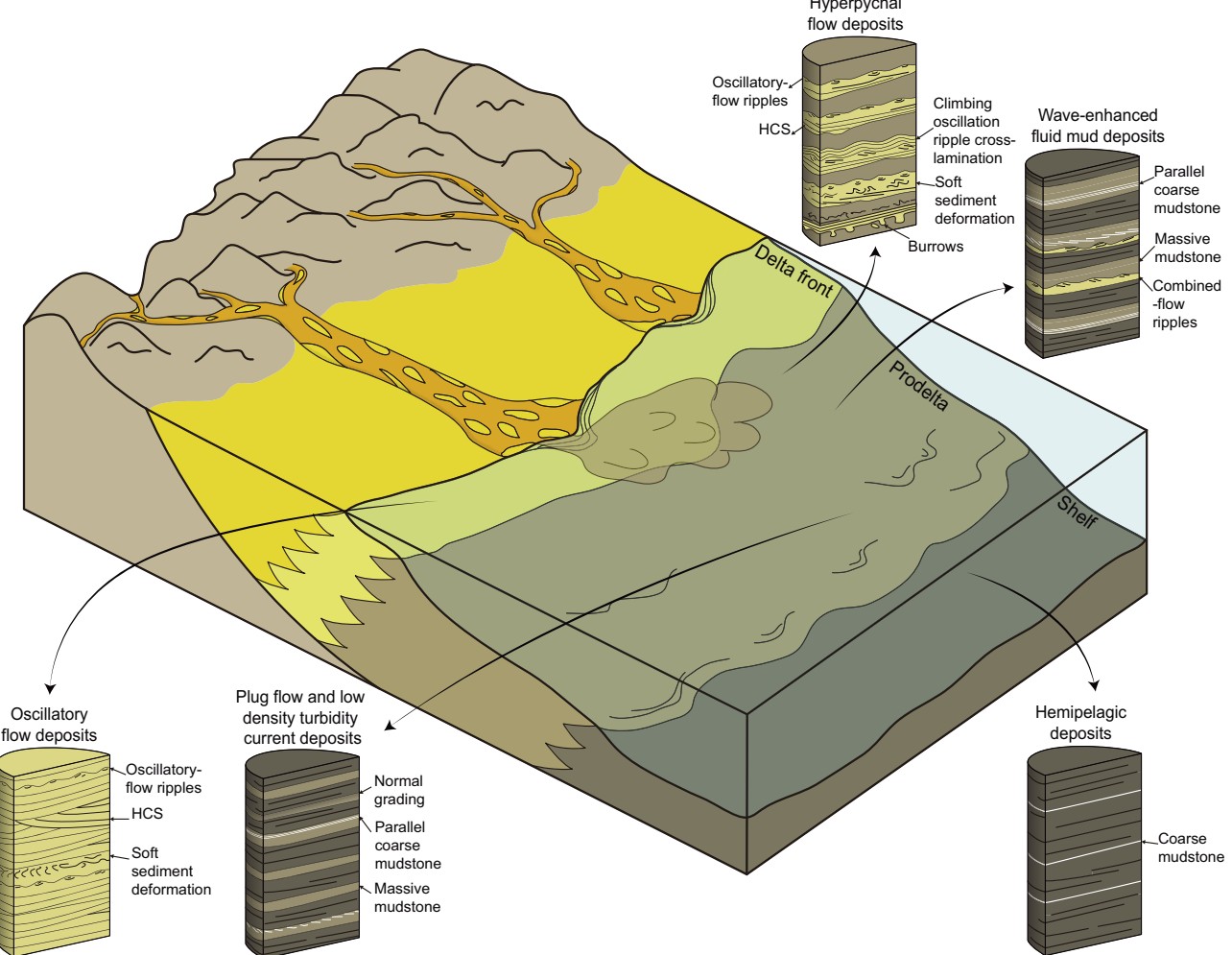

**Fig. 3 Block diagram showing the storm-flood-dominated delta with its characteristic deposits.** Idealized core intervals are shown for each type of deposit. Animals inhabited an oxygen-rich delta front and were transported by different types of flows to a more distal setting where preservation occurred under oxygen-depleted conditions. *HCS* hummocky cross-stratification.

and unidirectional currents during the waning stage. In addition to having been recorded in outcrops of the Yu'anshan Formation[28], gutter casts and scoured surfaces are extremely abundant in outcrops of similar deltaic systems elsewhere and are interpreted as resulting from erosion and deposition due to oscillatory-dominated combined flows[45]. These gutter casts are typically formed in subaqueous channels during storm-flood events[45–48]. Large and rapid sediment fluxes laying over an unconsolidated and unstable substrate are deemed responsible for soft-sediment deformation. Sands may have been introduced in the receiving basin during storm floods and subsequently reworked by oscillatory flows in a well-oxygenated, high-energy, shallow-marine environment above fair-weather wave base and in relative proximity to a river mouth. These deposits are interpreted as the most proximal facies recorded in the core (Fig. 3). Similar deposits have been observed elsewhere in comparable deltaic systems[45,46], typically representing proximal delta front environments.

Hyperpycnal flow deposits: These deposits consist of very thin- to medium-bedded, erosionally based, combined- to oscillatory-flow ripple cross-laminated, climbing oscillation-ripple cross-laminated, parallel-laminated and hummocky cross-stratified, light gray, well-sorted, very fine-grained sandstone, and mudstone (Fig. 2b–e, g–i, Supplementary Fig. 2a–n). Hummocky cross-stratification is of both anisotropic and isotropic types and

may be amalgamated locally. Although many sandstone beds show the vertical passage from hummocky cross-stratified or normally graded intervals to cross-laminated intervals, the opposite trend is apparent in other instances. Some of the associated mudstone interbeds may contain starved ripples (Supplementary Fig. 2a). Gutter casts are locally present (Supplementary Fig. 2e). Erosive basal surfaces are locally mantled by mudstone intraclasts. Soft-sediment deformation structures (e.g., convolute lamination, load casts) are common. Bioturbation degree is low to moderate (BI 0–4). Biodeformational structures delineating thin (<1.5 cm), bioturbated intervals are common in mudstone-dominated deposits. Some discrete trace fossils (e.g., small *Planolites* isp., *Palaeophycus* isp.) are common in sandstone-dominated deposits (Supplementary Fig. 2a). *Teichichnus rectus*, small *Rosselia* isp., *Bergaueria* isp., ? *Skolithos* isp., and escape trace fossils are present locally (Supplementary Fig. 2a–e, g, h). In addition, horizontal to rarely oblique, undetermined sandstone-filled burrows penetrating the underlying bioturbated mudstone from the base of sandstone beds also occur (Fig. 2f). These deposits are abundant in the Maotianshan Shale Member (Fig. 1). Evidence of oscillatory and combined flows is pervasive across the bedform spectrum (i.e., ripples and hummocks). Climbing oscillation-ripple cross-lamination results from the aggradation of oscillation ripples during combined wave reworking and high sedimentation rates[37].

Gutter casts provide further evidence of strong erosion. Vertical alternation of sedimentary structures suggests common fluctuations inflow velocity. Accordingly, these deposits are interpreted as the product of flood-derived, sustained hyperpycnal flows that occurred simultaneously with storms[28,40,45,47]. Alternatively, some of the beds with evidence of waning flows may represent turbidity flows associated with mouth bar collapse. In the case of the undetermined sandstone-filled burrows, which penetrate from the overlying sandstone, their characteristics (e.g., sharp unlined margins, passive sandstone fill from overlying layer, circular cross-section indicative of a lack of compaction after burial) suggest that they were emplaced into an erosionally exhumed firm substrate[49]. Thus, this bioturbation event took place after erosion and exhumation but prior to deposition of the overlying sand[50,51]. Firm substrates have been commonly recorded in Cambrian marine deposits and are interpreted to form owing to the absence of a soupy mixed layer[52,53]. However, the evidence that the firmground burrows of the Yu'anshan Formation commonly overprint a mottled background texture indicates that a previous softground trace-fossil suite was present. Therefore, we interpret that the firm substrates were generated due to erosional exhumation of a previously buried sediment rather than at the sediment-water interface. The presence of escape trace fossils provides additional evidence of rapid deposition. Sedimentologic characteristics suggest that these deposits were formed in distal delta front to proximal prodelta settings (Fig. 3) owing to increased wave energy and offshore-directed hyperpycnal and storm flows at times of peak fluvial discharge[45].

Wave-enhanced fluid mud deposits: These deposits comprise very thin- to thin-bedded, erosionally based, dark gray, homogeneous mudstone (Fig. 2e–g, Supplementary Fig. 3a–g). The erosive base is characterized by the marked scouring of the underlying sandstone deposits (Fig. 2e, f). In some cases, the top of the massive mudstone bed may display a transition to weakly developed parallel lamination marked by the occurrence of very thin coarse mudstone laminae. These deposits are typically unbioturbated (BI 0), but some undetermined burrows may occur, particularly in the laminated intervals (BI 1–3). These mudstone deposits invariably occur on top of hummocky cross-stratified and combined-ripple cross-laminated very fine-grained sandstone (Fig. 2e). These deposits are dominant in the Maotianshan Shale Member (Fig. 1). The mudstone beds are interpreted as produced by fluid muds, consisting of a bottom-hugging flow with a high concentration of clay and silt[38]. The erosive base is clearly inconsistent with sediment fallout and indicates emplacement by a sediment flow. The association with wave-generated structures suggests that these deposits may be linked to either storm wave resuspension of previously deposited mud or storm-enhanced fluvial runoff[38]. Conceptually, these mechanisms are akin to the wave-enhanced sediment gravity flows of Macquaker et al.[54] or the wave-modified turbidites of Myrow et al.[55]. The muddy wave-enhanced flow is a specific type of fluid mud deposit that occurs in distal delta fronts[38] to proximal prodeltas (Fig. 3).

Plug flow and low-density turbidity current deposits: These deposits comprise very thin- to thin-bedded, sharp- to erosionally based, light gray coarse mudstone alternating with dark gray fine mudstone intervals (Fig. 2g–j, Supplementary Fig. 4a–d). Coarse mudstone bases are flat or display small scour marks. These deposits are typically massive, although normal grading is observed in some layers (Fig. 2g, h, j). In some instances, coarse mudstone passes upwards into a parallel to current-ripple cross-laminated intervals, locally displaying continuous wavy parallel lamination, whereas other beds show upward transitions from parallel-laminated to inverse and normal-graded intervals. Diffuse internal boundaries are apparent as well in the coarse mudstone. Soft-sediment deformation structures, such as convolute lamination, are locally present. Fine mudstone intervals are mostly massive, but some may show subtle parallel lamination (Fig. 2j). Carbonate caps on coarse mudstone beds have been recorded[14]. The degree of bioturbation is low to moderate (BI 0-3) very locally. Small, shallow-tier mud-filled *Planolites* isp., in places forming clusters, and *Palaeophycus* isp. are the dominant ichnotaxa together with undifferentiated mottling (Supplementary Fig. 4a, b, d, e). Mantle and swirl structures are locally present (Supplementary Fig. 4d). These deposits are present mainly in the lowermost interval of the Maotianshan Shale Member (Fig. 1) and are the main host sediments for exceptionally preserved fossils. The overall characteristics of the coarse mudstone beds indicate deposition from unstable plug flow or quasi-laminar plug flows under high suspended-sediment concentration[31,56]. However, low-density turbidity currents may have been involved in the formation of the less abundant normally graded layers. Some of these currents may have been of surge-type triggered by delta front collapse. However, the local alternation of parallel-laminated and inversely and normally graded divisions with diffuse contacts suggests flow fluctuation. Accordingly, these intervals may have been produced by flood-derived, sustained hyperpycnal flows[40,47]. The local presence of wavy parallel lamination indicates oscillatory flows associated with storms, which may have reworked and enhanced the hyperpycnal flows[48]. The fine mudstone intervals may reflect in part deposition from the tail of an event flow and in part hemipelagic deposition. Either way, the scarcity of structures indicative of oscillatory flows indicates deposition very close to the storm wave base in a distal prodelta to mudbelt setting[37,57] (Fig. 3). The scarcity of bioturbation suggests that environmental conditions were unfavorable for most of the benthos. This is consistent with the interpretation that carbonate caps (thin beds cemented with carbonate) in these deposits were formed under low-oxygen bottom water conditions[14]. Local mantle and swirl structures suggest rapid colonization of muds after deposition in substrates with low consistency[58].

Hemipelagic deposits: These deposits consist of dark fine mudstone intervals of massive appearance (Fig. 2i, j, Supplementary Fig. 5a, b). These typically are thinly intercalated with other types of deposits (Fig. 2i, j), but in places can form thicker packages (up to >1 m) without disruption by event deposition (Fig. 1). Isolated silt and sand grains may occur sporadically within these intervals. Thin lamina rich in organic matter and carbonate is present locally. However, some coarse mudstone layers displaying parallel and current-ripple cross-lamination occur (Supplementary Fig. 5a). No bioturbation has been detected (BI 0) for the most part. However, in places where the sediment is more heterogeneous, the primary sedimentary fabric is locally disturbed, probably representing local biodeformational structures (BI 1–2, Supplementary Fig. 5a). These deposits are dominant in both the Black Siltstone Member and the Black Carbonaceous Member (Fig. 1). The overwhelming dominance of fine mudstone and the presence of organic matter-rich intervals suggest that thick packages of these deposits are the most distal ones of the whole succession (Fig. 3). Although low-energy, hemipelagic conditions are typically envisaged for these deposits, the local presence of cross-lamination suggests the participation of higher-energy tractive events[41]. The paucity of bioturbation suggests that low-oxygen conditions were predominant[59–61]. The distal-most areas of these muddy systems are typically characterized by sediment-starved shelf deposits enriched in organic matter[57].

**Depositional dynamics**. Our detailed description and analysis of the Yu'anshan Formation allow us to develop a depositional

model for the sediments that host the Chengjiang Biota. This depositional model provides a framework to understand the palaeoenvironmental setting where the animals lived, died, and were ultimately buried. This also facilitates comparisons of the Chengjiang Biota with fossil assemblages in other Cambrian Lagerstätten because the palaeoenvironment is likely to exert a strong control on faunal composition and paleoecology. Herein, we provide evidence that the Chengjiang Biota inhabited a deltaic environment influenced by storm floods.

Previous depositional models for the Chengjiang Biota have been contradictory, and a deltaic scenario has only been briefly mentioned without any supporting evidence[26–28]. Some studies invoked a shallow-water setting comparable to deposits of exceptional preservation in late Paleozoic tidally influenced estuarine environments[23,24]. However, no tidally generated structures have been detected in our sedimentologic analysis. Other studies envisaged deposition in foreshore-shoreface to offshore settings[21]. Wave-generated structures are abundant in outcrop[22,25] and are evident in cores as well (e.g., Fig. 2a). However, these are not the sole type of structures observed at the Yu'anshan Formation (Fig. 2), indicating that the environment was not only dominated by the action of wave processes operating along a strandplain. The association of hyperpycnal flows and wave structures suggests a proximity to a deltaic distributary channel mouth[28,45,62]. Delta front deposits may grade laterally into shoreface deposits, representing transitions to a strandplain along depositional strike as noted in many modern[45] and ancient[63,64] deltas. In these systems, there is a simultaneous response of rivers and coastal-shelf areas to storm events that affect both drainage and marine basins[35,48]. Strong winds and precipitation produced during storms result in river floods upstream and elevated wave heights downstream, triggering the formation of coupled storm-flood depositional systems[44,48,65]. In addition, the interaction between hyperpycnal flows and storm-generated wave reworking is conducive to remobilization, transport, and re-sedimentation of fine-grained material far away from the distributary channel mouth[66]. Thus, these depositional systems are referred to as storm-flood-dominated deltas[48] or hyperpycnal littoral deltas[66] (Fig. 3). Notably, geochemical analysis at Chengjiang indicates normal salinity during relatively dry fair-weather periods and reduced salinity during the wet season[22], further supporting discharge from distributary channels. The deltaic interpretation is in accordance with sedimentary characteristics observed in outcrops[25]. The studied core is located in a more proximal position than the outcrops that have been traditionally interpreted as wave-dominated shoreface to offshore and shelf deposits[25]. However, even in those outcrops, evidence of dilute turbidity flows has been recorded, underscoring the role of unidirectional tractive currents[25]. Sedimentological analysis of the Chengjiang succession supports the present view that the interplay of fluvial input and storms has been typically overlooked in the analyses of shallow-marine successions[48].

Most, if not all, studies of this type of deltas have been conducted in post-Paleozoic strata as well as in modern environments, such as in the Baram Delta of Borneo[45] and the Trent River of North Carolina[48]. However, the storm-flood model needs to be adjusted to account for the non-actualistic aspects of Cambrian sedimentation. For example, significant departures result from the contrasting nature of Cambrian drainage basins characterized by low-sinuosity sheet-braided-style fluvial systems in an alluvial landscape essentially devoid of vegetation cover[67] (Fig. 3). Cambrian alluvial plains were markedly deficient in mud due to the lack of fine-grained sediment retention mechanisms in the absence of plant cover[68]. As a result, sediment delivery to coastal and shelf areas was mostly episodic in nature and controlled by discharge variations in fluvial catchment zones[67].

This specific configuration of the Cambrian alluvial landscape may have augmented the effect of storms as triggers of flooding events. In addition, the palaeogeographic configuration (i.e., the South China microplate located at around 5–15°S) may have promoted the formation of relatively small "dirt" rivers prone to generate low-density hyperpycnal flows that can be easily deflected by longshore currents on low gradient shelves (Fig. 3)[37].

Ichnologic evidence also suggests deposition in a setting affected by periodic freshwater discharge rather than a fully marine environment only influenced by wave action. The overall low trace-fossil diversity, sparse and uneven distribution of bioturbation, the small size of bioturbators, presence of simple ichnofossil morphologies, the dominance of monospecific or paucispecific associations, and paucity of suspension-feeding burrows are all characteristics commonly recorded in deltaic successions[69]. It may be argued that in the case of the Yu'anshan Formation these ichnologic traits reflect evolutionary constraints rather than environmental factors, therefore invoking the anactualistic nature of the Cambrian to explain these features as trace fossils have displayed significant changes through the Phanerozoic[70,71]. However, the basic characteristics outlined above are best interpreted as reflecting the influence of environmental stressors associated with a nearby fluvial source. Whilst the intensity and depth of bioturbation have experienced a marked increase through the Phanerozoic[71–74], pervasively bioturbated deposits under normal marine conditions are known since the early Cambrian[75,76]. Several Lower Cambrian fully marine deposits are characterized by a high diversity of trace fossils, commonly including large sizes and complex morphologies, reflecting colonization by a diverse infauna under relatively stable conditions[77,78]. As with younger deposits, monospecific suites are typically associated with various stressors (e.g., brackish-water, dysoxia) regardless of their age. Therefore, the pattern of sparse and uneven bioturbation we describe in the Yu'anshan Formation is more consistent with environmental stressors, such as high sedimentation rates and freshwater discharge, resulting from the proximity of the deltaic mouth. In the same vein, suspension feeders are typically inhibited in deltaic settings due to water turbidity that tends to clog filtering devices[69]. The marked scarcity of dwelling burrows of suspension feeders in the Yu'anshan Formation can hardly be attributed to evolutionary constraints, as these structures are common in fully marine deposits of the same age[73].

**Taphonomic conditions.** Carcasses of the Chengjiang Biota could have become preserved in sediments deposited in the delta front and the shelf settings but the most exquisitely preserved fossils are associated with plug flows and low-density turbidity deposits in the prodelta (Fig. 1). The prodelta environment is an ideal setting for exceptional preservation to occur because it is an area prone to rapid event-bed deposition (i.e., unlike the shelf that records mainly hemipelagic deposits; Fig. 3) such that carcasses were rapidly buried protecting them from macro-scavengers. The relatively low-energy environment, compared with the delta front (Fig. 3), and fine-grained lithology may also have facilitated preservation. Evidence, such as limited bioturbation, the abundance, and distribution of redox elements[18], and carbonate cemented bed tops[14], indicates that the prodelta sediments alternated between dysoxic and anoxic conditions (see Supplementary Table 1 for Oxygenation data)—a characteristic linked to exceptional preservation across many Cambrian Lagerstätten[79]. The lack of oxygen in the prodelta suggests that most animals must have been living in shallower oxygenated waters and were transported by flows from the delta front to the prodelta where preservation occurred. The high degree of preservation associated

with plug flows in the prodelta (Fig. 1) can result as well from the dampened turbulence of these flows[80].

**Paleoecological implications**. From a paleoecological standpoint, the Chengjiang Biota inhabited a well-oxygenated, nutrient-rich nearshore deltaic environment (see Supplementary Table 1 for Oxygenation data). Oxygen and nutrients are abundant in shallow deltaic settings as the water column is easily agitated by wave processes, and rivers typically transport a large array of chemical compounds to the delta[69]. Currently, deltaic environments have not been generally inferred for any other Cambrian deposit with the soft-tissue preservation[79], although a fan delta setting has been proposed for the Emu Bay Shale[81].

Even though oxygen and nutrients are abundant in the delta front, environmental conditions must have been unstable and characterized by strong fluctuations, including significant seasonal freshwater discharge and high sedimentation rates[69,82,83]. In the Chengjiang Biota, a recently discovered fossil site yielded a large number of juveniles[3]. This accords with a deltaic setting and high sediment fluxes which can bury sessile organisms, preventing them from attaining adult stage[84,85]. High particle suspension in shallow settings of deltaic environments can prohibit the growth of filter-feeding taxa in these settings as well (i.e., clogging effect). Thus, filter feeders are either absent from shallow deltaic settings[69] or tend to be rare and small in size[3]. This observation may explain the in-situ preservation of normal-sized filter-feeding sponges in distal-most environments of the Chengjiang Biota, unlike most other phyla that have been transported from shallower-water settings[18]. Furthermore, freshwater input[22] may have been responsible for mass mortality events[86,87]. For example, in the Chengjiang Biota there are bedding planes where hundreds of specimens of *Yunnanozoon* are preserved exhibiting minimal decay[88]. These assemblages may represent intervals where salinity fluctuations occurred as a result of freshwater discharge. Moreover, fluctuating salinity can explain why echinoderms, one of few stenohaline phyla in the animal kingdom[89], are absent from the Chengjiang Biota but are present (even if in low numbers) in other Burgess Shale-type open marine settings like those of the Walcott Quarry[8] and are abundant in sites such as the Kaili Biota[90].

In summary, the Chengjiang Biota inhabited a storm-flood-dominated delta. This is the only Burgess Shale-type deposit to be associated with such a deltaic environment. This shallow-marine environment was unstable and characterized by fluctuating salinity and high sedimentation rates, explaining the mass mortalities of *Yunnanozoon*[88], the abundance of juveniles in specific localities[3], and the absence of stenohaline echinoderms while questioning the long-held view of Burgess Shale-type faunas as snapshots of stable, more distal shelf and slope communities. Application of recent developments in the field of mudstone sedimentary dynamics to Burgess Shale-type deposits may revolutionize our understanding of the ecology and taphonomy of these exceptional deposits.

## Methods

**Core sampling**. A 130 m-thick core was drilled in Jinning County (24°42′59″N, 102°31′09″E), Eastern Yunnan Province, China. This core spans the entire Yu'anshan Formation, including the Maotianshan Shale Member, in which the Chengjiang Biota was discovered. During the Cambrian, this area was situated at the southwestern edge of the Yangtze Platform and was connected with the open ocean as the sea deepened gradually from west to east. Half of the core was sampled for geochemical analyses, and half of it is deposited in the sedimentary archive of Yunnan Key Laboratory for Palaeobiology at the Chenggong Campus of Yunnan University, China. The archived half was used for this study's non-destructive sedimentary facies analyses of this study and can be accessed freely upon request. We prepared 161 thin sections (30 μm thick), typically of 5.5 cm × 3.0 cm (with the largest ones being 7.0 cm × 3.5 cm).

**Bioturbation and trace-fossil classification**. The degree of bioturbation is based on the BI, which comprises a scale from zero to six. BI is equal to zero if bioturbation is absent. BI = 1 if the percentage of bioturbation is between 1 and 4% with distinct bedding, few discrete traces, and/or escape structures. BI = 2 if bioturbation percentage is between 5 and 30% with low trace density and common escape structures. BI = 3 if between 31 and 60% of the sediment is bioturbated with a rare overlap of traces. BI = 4 if bioturbation is high (61–90%) with a common overlap of traces and primary sedimentary structures are mostly erased. BI = 5 is characterized by intense bioturbation (91–99%) and sediment with almost completely disturbed bedding. BI = 6 when the sediment is fully bioturbated. Trace fossils were classified following conventional practices in ichnotaxonomy. The term "isp." is used as an abbreviation of ichnospecies and employed in those cases in which assignment was done at ichnogenus level. Occurrences in which all the diagnostic features of the corresponding ichnotaxon cannot be confirmed were noted with the "?" sign.

**Palaeoenvironmental terminology standardization**. Core descriptions were supplemented with analysis of thin sections and outcrop observations at the Xiaolantian and Kunyang Phosphate Mine sections. The palaeoenvironmental terminology used in the Chengjiang literature has been somewhat inconsistent and, therefore, is standardized in this study. For environmental subdivisions, we consider the shoreface as the region between the low tide line and the fair-weather wave base, the offshore as the area between fair-weather wave base and storm wave base, and the shelf as lying between storm wave base and the slope break. The deltaic environment is subdivided following standard practices into delta plain, delta front, and prodelta.

## Data availability

The data generated in this study are provided in the Supplementary Information and Main Manuscript file.

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

## Acknowledgements

This paper is supported by grant no. 2020M683388 from the Chinese Postdoctoral Science Foundation awarded to F.S.; by grant no. 41763002 from NSFC and grant no. 113113 from the State Key Laboratory of Palaeobiology and Stratigraphy (Nanjing Institute of Geology and Paleontology and Center for Excellence in Life and Palaeoenvironment, CAS) awarded to C.Q.; by Natural Sciences and Engineering Research Council (NSERC) Discovery Grants 311727–20 and 422931-20 awarded to M.G.M, and L.A.B. M.G.M. thanks additional funding by the George J. McLeod Enhancement Chair in Geology.

## Author contributions

F.S., C.Q., and X.M. designed the research. X.H. and C.Q. secured the funding and retrieved the cores. C.Q. and Q.Z. took all photos and compiled the data. C.Q., S.E.G., and X.M. initiated the geological background study on the core and correlated the fossil outcrops with the core. F.S., L.A.B., M.G.M., M.P., and R.V. described and interpreted the sedimentary facies and wrote the paper. All authors participated in the discussions and edits to the manuscript.

## Competing interests

The authors declare no competing interests.
