## [Peer Review File · Nature Communications]

The Chengjiang Biota inhabited a deltaic environmentREVIEWER COMMENTS

Reviewer #1 (Remarks to the Author):

Review of "The Chengjiang Biota inhabited a deltaic environment" by Saleh et al., provided by Julien Kimmig

Dear Farid and colleagues,

First, let me say congratulations on the very thorough and well written study. It is great to see that work is being done on the interpretation of depositional environments of Cambrian Lagerstätten and this study will be very important moving forward. I think it is a great fit for Nature Communications.

While I have some minor comments on the text and some of the interpretations the major thing, I would like to see are more pictures of the sedimentary structures that are mentioned. This could be in the main article or the supplement, as they will support the interpretations presented in the article.

- What are the noteworthy results?

The well supported interpretation of the Chengjiang biota having lived in a deltaic environment rather than an open marine setting is very important to future studies not only of the Chengjiang biota, but also other sites preserving exceptional fossils through time and space.

- Will the work be of significance to the field and related fields? How does it compare to the established literature? If the work is not original, please provide relevant references.

Yes, this work will be very important to the understanding of depositional environments of marine Lagerstätten, similar sedimentary deposits and is a fantastic step forward in the understanding of Cambrian paleoenvironments.

- Does the work support the conclusions and claims, or is additional evidence needed?

The text supports the conclusions and claims, but it would be nice to have additional photographs of some of the mentioned sediment structures at least in the supplementary material. Additionally I pointed out a couple of areas where I think the interpretations might need some 'mellowing' or adjustments.

- Are there any flaws in the data analysis, interpretation and conclusions?

The idea of the Chengjiang biota inhabiting a deltaic environment was mentioned before, alas with little support (Hu 2005 Taphonomy and Palaeoecology of the Early Cambrian Chengjiang Biota from Eastern Yunnan, China and references therein), and it should likely be mentioned in the current paper.

- Do these prohibit publication or require revision?

Does not prohibit publication, but should likely be mentioned.

- Is the methodology sound?

The methodology is appropriate, but some additional information needs to be provided. How many thin sections were prepared? What size were the thin sections, and how thick were they? Which outcrops were visited for 'outcrop observations'?

Does the work meet the expected standards in your field?

Yes

- Is there enough detail provided in the methods for the work to be reproduced?

See comment above.

Minor Comments:

Line 28: maybe change to 'The Chengjiang Biota is one of the earliest Cambrian soft-bodied fossil assemblages', considering Ediacaran biotas

Line 44: As you mention Yang et al. 2021 later, it fits well here as it lists one of the more up to date taxa lists of the Chengjiang biota

Line 60: Lagerstätten not Lagersätten

Line 62: Lagerstätten not Lagersttäten

Line 70: I agree it is very rare, but some attempts, if by far not as detailed as this one, and on outcrop samples, have been performed in the last decade (i.e., Garson et al. 2012 Lethaia Spence Shale, Kimmig and Pratt 2016 Lethaia Ravens Throat River, Kimmig et al. 2019 JGS Box. 1 Spence Shale).

Line 111: Would it be possible to add a picture of the interbedded very thin mudstones in the supplement?

Line 111: It would also be great to figure the soft-sediment deformation structures.

Line 131: It would also be great to figure the scour surfaces and gutter casts so other researchers know what to look for if they can be hard to identify in core.

Line 152: Escape traces instead of 'Escape trace fossil'?

Line 152: It would also be great to figure one of the escape traces.

Line 183: Were there any obvious carbonate caps in this core?

Line 190: It would also be great to figure one of the escape traces, as they support your interpretation.

Line 212: It would also be great to figure one of the cross-lamination areas, as your interpretation is based on it.

Line 219: It would be good to mention which deposits were investigated.

Line 226: Hu 2005 (Taphonomy and Palaeoecology of the Early Cambrian Chengjiang Biota from Eastern Yunnan, China) and references therein mention the possibility of the deltaic environment, even so it is mostly based on poor evidence or contradicting the possibility. But this should likely be mentioned.

Line 245-246: While this is one interpretation of the data, the Zhu et al. (2001) did not correct for enrichment factors, and also did not solidify their data with any isotopic analyses, as such it is to be taken with a grain of salt.

Line 280: The sediments being dysoxic to anoxic does not necessarily mean that the water column was dysoxic to anoxic. While transportation is always a possibility, there are some of the exceptional Chengjiang deposits that preserve burrows, and burrowing animals, and as such likely represent in-situ deposits, while others have been transported.

Line 291: See above comment Hu 2005.

Line 300: There are several filter feeding species, including Herpetogaster, present at the Haiyan locality (Yang et al. 2020 Science of Nature), which going back to the dysoxic to anoxic environments, and the present of the juveniles at this locality, likely fits the special setting here. Do you see some way to fit this in your model?

Line 302: Saying all other phyla have been transported might be extensive, maybe 'most other specimens/phyla'?

Line 305: 'are' not in italics

Reviewer #2 (Remarks to the Author):

I have reviewed the submission by Saleh et al. on the interpretation of the Yu'an-shan Formation that hosts the soft-bodied Chengjiang biota as broadly storm-flood prodeltaic. The paper is quite well written and illustrated. It presents a reasonably robust taphonomic reinterpretation of the host rocks that integrates detailed interpretation of physical sedimentary structures and ichnology to bolster the interpretation of the environments as rather shallower than previously interpreted.

I think the paper is publishable with minor revisions.

1. It would be good to comment that Cambrian ichnology is rather different from Ordovician to modern when more complex and deeper tiering developed as well as increasing diversity associated with GOBE. Is it fair to think of the environment as stressed as if this was a Cretaceous example? I agree that the burrowing all looks a little sparse, but there also the behaviours and animals that show more complex burrowing behaviours had not evolved in the early Cambrian so the non-actualistic nature of this time should require some caution in using ichnology models based on times of higher diversity. Also, algal grazing may not have been as robust in the early Cambrian. Could any of these fluid-muds be stabilized by a mat-ground? I agree that none of these photos show stromatolites, but some comment on substrate strength may be worth noting. A lot of the burrows look like firmground suites which seems less compatible with rapidly deposited porosity rich muds. Some comment on whether the substrate is also non-actualistic might be warranted and may help understand the complex taphonomy of this time.

2. In Figure 1, I would emphasize coarsening upward, which is a common motif of prograding shallow marine depositional systems. Also, the top of the core says it's the Upper Siltstone Member but isn't more of a sandstone as shown in Fig. 2A?

3. Although I like hyperpycnites, the key evidence is inverse grading. In 2E, for example, I see lots of normal grading and Bouma sequences. Why aren't these just regular turbidites. It may be a bit semantic, but a hyperpycnite usually implies a sustained flow that waxes and wanes versus turbidity currents that surge and wane. Delta front turbidites may form by mouth bar collapse and could be linked to river floods. To that end, it is not well argued as to why the climbing ripples sandstones must be hyperpycnites versus waning flood layers? Is it because they appear to be sustained? Is there evidence of waxing flow? This is usually critical to convincingly argue for hyperpycnal flows versus surging-waning turbidity currents. Having turbidites doesn't detract from the overall story, especially with the observation of upward coarsening and wave-formed structures.

4. It would be nice to show the bioturbation index to illustrate the sporadic distribution of the trace fossils. Do you see any navichnia (mantle and swirl)? Please could you label the key trace fossils and talk about their environmental significance (of course emphasizing the reduced diversity of Cambrian trace makers and ethologies).

1. What are the noteworthy results?

Yes. Paper revises interpretation of depositional environments of the Chengjiang lagerstätten and may have implications for other soft bodies fossil sites.

2. Will the work be of significance to the field and related fields? How does it compare to the established literature? If the work is not original, please provide relevant references.

Good application of sedimentology and ichnology but could use a bit more specifics on ichnology (BI, name trace fossils) and comment that Cambrian ichnology may be non-actualistic.

3. Does the work support the conclusions and claims, or is additional evidence needed?

Could use a bit more clarity on interpretation of hyperpycnites, These may be over interpreted here.

4. Are there any flaws in the data analysis, interpretation and conclusions? Do these prohibit publication or require revision?

See comments...

5. Is the methodology sound? Does the work meet the expected standards in your field?

See comment, could use a bit more systematic description of trace fossils

6. Is there enough detail provided in the methods for the work to be reproduced?

yes

Dr. Janok P. Bhattacharya
Susan Cunningham Research Chair in Geology
School of Earth, Environment and Society (SEES)
McMaster University

Reviewer #3 (Remarks to the Author):

This is a well-written, well-argued and informative paper, which presents detailed palaeoenvironmental context for a key fossil site. It should be of broad general interest and is scientifically sound. I have only a few minor comments (below) and would like to see this published:

p. 2, l. 33 – ‘storm floods’ is an ambiguous term, and it is unclear if this means fluvial or marine flooding. The details are explained in the main text but the wording here in the abstract could be more informative.

p. 3, l. 47 – “Stage 5 “ should now be Wuliuan.

p. 5, l. 108 – it should be nearly impossible to undisputedly identify HCS in a core, lacking the 3D aspect of this structure. This should be reworded to a passive description of the laminae, and emphasise that it is *interpreted* as HCS

p. 10, l. 229 – as many tidal signatures can be cryptic, depending on tidal range and asymmetry, it is important to note that absence of evidence is very much not the same as evidence for absence when it comes to tidal sedimentation

p. 13, l. 299 – could this nearshore area effectively be a nursery for juveniles of open marine species?

We are pleased to see that all reviewers are very positive about our manuscript. Below, we and respond to every comment and suggestion point by point. All changes in the MS are marked in blue text. We would like to thank the Reviewers who provided excellent and rigorous feedback. We feel the manuscript is now further improved, and we are hopeful you will consider it for publication.

Response to Reviewers

Reviewer #1: Julien Kimmig

First, let me say congratulations on the very thorough and well written study. It is great to see that work is being done on the interpretation of depositional environments of Cambrian Lagerstätten and this study will be very important moving forward. I think it is a great fit for Nature Communications.

We thank the Reviewer for his positive comments.

While I have some minor comments on the text and some of the interpretations the major thing, I would like to see are more pictures of the sedimentary structures that are mentioned. This could be in the main article or the supplement, as they will support the interpretations presented in the article.

We thank the Reviewer for this comment as it gives us the opportunity to add images from core sediments and their related thin sections, which we think fully illustrate the features, textures and structures in the deposit and allow readers to fully appreciate our interpretations. There are now five supplementary figures in the new version, which illustrate all the main features we describe and interpret.

- What are the noteworthy results?

The well supported interpretation of the Chengjiang biota having lived in a deltaic environment rather than an open marine setting is very important to future studies not only of the Chengjiang biota, but also other sites preserving exceptional fossils through time and space.

Thank you for agreeing on the significance of this work.

- Will the work be of significance to the field and related fields? How does it compare to the established literature? If the work is not original, please provide relevant references. Yes, this work will be very important to the understanding of depositional environments of marine Lagerstätten, similar sedimentary deposits and is a fantastic step forward in the understanding of Cambrian paleoenvironments.

Thank you.

- Does the work support the conclusions and claims, or is additional evidence needed?

The text supports the conclusions and claims, but it would be nice to have additional photographs of some of the mentioned sediment structures at least in the supplementary material. Additionally I pointed out a couple of areas where I think the interpretations might need some 'mellowing' or adjustments.

In the revised version, we have added core photos that show the most important features of the deposits studied, and we have toned down some of the statements when suggested by Reviewer 1 (see their comment entitled "line 70" below).

- Are there any flaws in the data analysis, interpretation and conclusions?

The idea of the Chengjiang biota inhabiting a deltaic environment was mentioned before, alas with little support (Hu 2005 Taphonomy and Palaeoecology of the Early Cambrian Chengjiang Biota from Eastern Yunnan, China and references therein), and it should likely be mentioned in the current paper.

Hu's 2005 thesis does mention that a deltaic environment could be a possibility for the Chengjiang biota, but the author concludes "The delta environment as suggested by some is not consistent with the sedimentary structures, etc..." (p 27). Therefore, we think that our statement stands "Despite hundreds of research articles describing taxa from the Chengjiang Biota⁹⁻¹² and their preservation¹³⁻²⁰, there is no consensus interpretation on the paleoenvironment in which animals lived, died and were preserved. A variety of depositional environments have been invoked ranging from tidally influenced estuaries and embayments, to wave-dominated shoreface-offshore complexes and shelf settings affected by turbidity currents²¹⁻²⁸." We think this is a fair reflection of the state of the knowledge on Chengjiang depositional environment(s) before our work herein. However, we have included three more citations, including that of Hu et al., within this paragraph to be more inclusive.

- Do these prohibit publication or require revision?

Does not prohibit publication, but should likely be mentioned.

We have added the reference of Hu (2005) (see above).

- Is the methodology sound?

The methodology is appropriate, but some additional information needs to be provided. How many thin sections were prepared? What size were the thin sections, and how thick were they? Which outcrops were visited for 'outcrop observations'?

The requested information has been added (lines 91-92 & 96). We have now specified the number and characteristics of the prepared thin sections and the names of the outcrops studied.

Does the work meet the expected standards in your field?

Yes

- Is there enough detail provided in the methods for the work to be reproduced?

See comment above.

Minor Comments:

Line 28: maybe change to 'The Chengjiang Biota is one of the earliest Cambrian soft-bodied fossil assemblages', considering Ediacaran biotas

We have rephrased the sentence following the Reviewer comments (although we use "Phanerozoic" instead of "Cambrian" to avoid using the later twice in the same sentence).

Line 44: As you mention Yang et al. 2021 later, it fits well here as it lists one of the more up to date taxa lists of the Chengjiang biota

We have added this.

Line 60: Lagerstätten not Lagersätten

Corrected.

Line 62: Lagerstätten not Lagerstäten

Corrected.

Line 70: I agree it is very rare, but some attempts, if by far not as detailed as this one, and on outcrop samples, have been performed in the last decade (i.e., Garson et al. 2012 Lethaia Spence Shale, Kimmig and Pratt 2016 Lethaia Ravens Throat River, Kimmig et al. 2019 JGS Box. 1 Spence Shale).

Although all remarkable in their own right, the studies mentioned do not have the level of sedimentologic detail that has been adopted in mudstone sedimentology after the paradigm shift. However, we have toned down our statement by stating that “has rarely been applied” instead of “has not been applied”.

Line 111: Would it be possible to add a picture of the interbedded very thin mudstones in the supplement?

Added in Suppl. Material 1A and D.

Line 111: It would also be great to figure the soft-sediment deformation structures.

Added in Suppl. Material 1A and C.

Line 131: It would also be great to figure the scour surfaces and gutter casts so other researchers know what to look for if they can be hard to identify in core.

Added in Suppl. Material 1D. This structure is now part of the oscillatory flow deposits, as it is mantled by HCS.

Line 152: Escape traces instead of ‘Escape trace fossil’?

We follow here the convention of using “trace fossils” when dealing with fossil material.

Line 152: It would also be great to figure one of the escape traces.

Added in Suppl. Material 2D, G.

Line 183: Were there any obvious carbonate caps in this core?

Although very rare, carbonate caps can be found. They consist of very thin levels (less than a couple of mm-thick). Please note that this observation is not contradictory with previous findings of abundant carbonate caps associated with exceptional fossil preservation (Gaines et al., 2012). Gaines et al. (2012), geochemically investigated a different core from Chengjiang. Thus, it is expected to have some differences between both (more carbonate caps in Gaines et al., 2012). In this manuscript, we discuss the results of Gaines et al. (2012) by putting them in the broad scheme of the sedimentary environment advanced herein.

Line 190: It would also be great to figure one of the escape traces, as they support your interpretation.

They are now figured in Suppl. Material (Supplementary Figure 2B, D, G, H).

Line 212: It would also be great to figure one of the cross-lamination areas, as your interpretation is based on it.

Added in Suppl. Material 5A.

Line 219: It would be good to mention which deposits were investigated.

Added in line 96.

Line 226: Hu 2005 (Taphonomy and Palaeoecology of the Early Cambrian Chengjiang Biota from Eastern Yunnan, China) and references therein mention the possibility of the deltaic environment, even so it is mostly based on poor evidence or contradicting the possibility. But this should likely be mentioned.

We thank the Reviewer for pointing to this omission; we are citing this study in the revised version.

Line 245-246: While this is one interpretation of the data, the Zhu et al. (2001) did not correct for enrichment factors, and also did not solidify their data with any isotopic analyses, as such it is to be taken with a grain of salt.

We fully agree, so we are just pointing that this is consistent with our interpretation, but without elaborating much more based on this evidence.

Line 280: The sediments being dysoxic to anoxic does not necessarily mean that the water column was dysoxic to anoxic. While transportation is always a possibility, there are some of the exceptional Chengjiang deposits that preserve burrows, and burrowing animals, and as such likely represent in-situ deposits, while others have been transported.

It is seen in figure 1 and mentioned in lines 324-326 in the manuscript that exceptional fossil preservation can occur in a wide range of environments within the delta. However, the quality of preservation differs between different deltaic deposits. For instance, most of the “best” exceptional preservation is associated with plug flow and low-density turbidity current deposit, and these animals were likely transported from shallower-water settings. However, Figure 1 shows that some exceptional fossil preservation (lower degree of fidelity) occurred within either shallower- or deeper-water settings, and some of these, particularly in the deeper-water setting, might have not been transported. In the manuscript, it is mentioned that: “This observation explains the in-situ preservation of filter-feeding sponges in distal environments of the Chengjiang Biota, unlike most other phyla that have been transported from shallower settings” (lines 355-358). Regarding oxygenation, we fully agree with Reviewer 1 as well. In fact, even if the sediment is anoxic or dysoxic, the water column can have considerable amounts of oxygen. That is why upon transport of material and the establishment of anoxic-dysoxic conditions within the sediments, the top of these sediments can get superficially bioturbated. Shallow-tier bioturbation is consistent with the presence of a very shallow redox discontinuity surface. However, to make these aspects clearer in the new version of Figure 1, we have added the bioturbation index as per the recommendation of Reviewer 2. As a result, this issue raised by Reviewer 1 is now addressed, as it is clearly seen that some bioturbation (very minor) exists within the interval of “best” exceptional preservation. Also, kindly note, that within the manuscript the term of anoxic/dysoxic were associated to sediments only and were not used to describe the water column.

Line 291: See above comment Hu 2005.

We have added the reference to Hu 2005 in the manuscript.

Line 300: There are several filter feeding species, including *Herpetogaster*, present at the Haiyan locality (Yang et al. 2020 Science of Nature), which going back to the dysoxic to anoxic environments, and the present of the juveniles at this locality, likely fits the special setting here. Do you see some way to fit this in your model?

The Haiyan locality is definitely a very interesting locality. We think that the answer to the “weird” animal distribution in Haiyan is not the result of geochemical oxygen conditions but more likely the result of its particular sedimentary setting. Primary lithological data presented within Yang et al. (2021) in Nature Ecology and Evolution (cited in our manuscript) suggests that Haiyan is silt- and sand-dominated, with around 20% of mud. A preliminary comparison with the section presented in this manuscript (Fig. 1) indicates that Haiyan was generally more proximal. Of course, it would be extremely helpful if a core was available at Haiyan to confirm this pattern.

It is also worth noting that Reviewer 3 thinks of Haiyan as a nearshore setting too (as pointed in their last comment). The most probable proximal setting of Haiyan can explain the distribution of animals in this locality. Although the proximal setting of Haiyan must have been extremely rich in nutrients and oxygen, it must have also been filled with particles in suspension in the water column. Filter feeding animals such as *Herpetogaster* are negatively impacted by the silt and clay in suspension in the water column because these particles have a clogging effect. This is one of the reasons that can explain why animals did not reach larger sizes in Haiyan - they are dominantly preserved as juveniles. Of course, the abundance of physical disturbances, salinity fluctuations, etc... are other parameters that can be responsible for the abundance of juveniles in proximal sites. Note that similar processes have been suggested to explain body-size fluctuations in other sites with soft-tissue preservation, such as the Fezouata Biota (e.g., Saleh et al., 2018 in Palaios, and 2021 in Geol Mag). In order to accommodate for this explanation, without making any strong statement on Haiyan (prior to the realization of a core that investigates in detail that locality), we added a couple of sentences in the discussion (lines 353-355) to answer this comment by Reviewer 1. “High particle suspension in shallow settings of deltaic environments can prohibit the growth of filter-feeding taxa in these settings as well (i.e., clogging effect). Thus, filter feeders are either absent from shallow deltaic settings⁷³ or tend to be rare and small in size³.”

Line 302: Saying all other phyla have been transported might be extensive, maybe ‘most other specimens/phyla’?

Done.

Line 305: ‘are’ not in italics

Corrected

Reviewer #2: Janok P. Bhattacharya

I have reviewed the submission by Saleh et al. on the interpretation of the Yu’anshan Formation that hosts the soft-bodied Chengjiang biota as broadly storm-flood prodeltaic. The paper is quite well written and illustrated. It presents a reasonably robust taphonomic reinterpretation of the host rocks that integrates detailed interpretation of physical sedimentary structures and ichnology to bolster the interpretation of the environments as rather shallower than previously interpreted. I think the paper is publishable with minor revisions.

We thank the Reviewer for his positive comments.

1. It would be good to comment that Cambrian ichnology is rather different from Ordovician to modern when more complex and deeper tiering developed as well as increasing diversity associated with GOBE. Is it fair to think of the environment as stressed as if this was a Cretaceous example? I agree that the burrowing all looks a little sparse, but there also the behaviours and animals that show more complex burrowing behaviours had not evolved in the early Cambrian so the non-actualistic nature of this time should require some caution in using ichnology models based on times of higher diversity. Also, algal grazing may not have been as robust in the early Cambrian. Could any of these fluid-muds be stabilized by a mat-ground? I agree that none of these photos show stromatolites, but some comment on substrate strength may be worth noting. A lot of the burrows look like firmground suites which seems less compatible with rapidly deposited porosity rich muds. Some comment on whether the substrate is also non-actualistic might be warranted and may help understand the complex taphonomy of this time.

The Reviewer brings an excellent point that we are happy to address in this revised version. We have added a paragraph in which we explain the rationale to explain the ichnologic characteristics of these deposits in terms of environmental stressors, rather than evolutionary constraints (lines 298-321). We tried to keep our arguments short and straight to the point, but we would be happy to make further additions if needed.

We have added: "Ichnologic evidence also suggests deposition in a setting affected by periodic freshwater discharge rather than a fully marine environment only influenced by wave action. The overall low trace-fossil diversity, sparse and uneven distribution of bioturbation, small size of bioturbators, presence of simple ichnofossil morphologies, dominance of monospecific or paucispecific associations, and paucity of suspension-feeding burrows are all characteristics commonly recorded in deltaic successions⁷³. It may be argued that in the case of the Yu'an-shan Formation, these ichnologic traits reflect evolutionary constraints rather than environmental factors, therefore invoking the anactualistic nature of the Cambrian to explain these features as trace fossils have displayed significant changes through the Phanerozoic^{74, 75}. However, the basic characteristics outlined above are best interpreted as reflecting the influence of environmental stressors associated with a nearby fluvial source. Whilst the intensity and depth of bioturbation have experienced a marked increase through the Phanerozoic⁷⁵⁻⁷⁸, pervasively bioturbated deposits under normal marine conditions are known since the early Cambrian^{79,80}. Several Lower Cambrian fully marine deposits are characterized by a high diversity of trace fossils, commonly including large sizes and complex morphologies, reflecting colonization by a diverse infauna under relatively stable conditions^{81, 82}. As with younger deposits, monospecific suites are typically associated with various stressors (e.g., brackish-water, dysoxia) regardless of their age. Therefore, the pattern of sparse and uneven bioturbation we describe in the Yuan'shan Formation is more consistent with environmental stressors, such as high sedimentation rates and freshwater discharge, resulting from the proximity of the deltaic mouth. In the same vein, suspension feeders are typically inhibited in deltaic settings due to water turbidity that tends to clog filtering devices⁷³. The lack of dwelling burrows of suspension feeders in the Yu'an-shan Formation can hardly be attributed to evolutionary constraints, as these structures are common in fully marine deposits of the same age⁷⁷."

2. In Figure 1, I would emphasize coarsening upward, which is a common motif of prograding

shallow marine depositional systems. Also, the top of the core says it's the Upper Siltstone Member but isn't more of a sandstone as shown in Fig. 2A?

We have addressed this point by the addition of the following sentences in the figure caption "*Note the clear large-scale coarsening- and thickening-upward trend that is common in wave-influenced shallow-marine depositional systems, including most notably deltas. The so-called "Upper Siltstone Member" is actually sandstone-dominated.*" We have to keep the established lithostratigraphic nomenclature, although we made a point of clarifying that the term "Upper Siltstone Member" is misleading in the caption of figure 1.

3. Although I like hyperpycnites, the key evidence is inverse grading. In 2E, for example, I see lots of normal grading and Bouma sequences. Why aren't these just regular turbidites. It may be a bit semantic, but a hyperpycnite usually implies a sustained flow that waxes and wanes versus turbidity currents that surge and wane. Delta front turbidites may form by mouth bar collapse and could be linked to river floods. To that end, it is not well argued as to why the climbing ripples sandstones must be hyperpycnites versus waning flood layers? Is it because they appear to be sustained? Is there evidence of waxing flow? This is usually critical to convincingly argue for hyperpycnal flows versus surging-waning turbidity currents. Having turbidites doesn't detract from the overall story, especially with the observation of upward coarsening and wave-formed structures.

Agreed and thanks to the Reviewer for these insights. We have now added the possibility of delta-front turbidity flows in the description of the "Hyperpycnal flow" deposits (lines 159-160) to accommodate those deposits in which evidence of flow fluctuation is not apparent. However, we should highlight that we have inversely graded beds in Fig. 2G and I, and Supplementary Material 3F and 4B, providing evidence of a waxing flow. Further evidence of waxing flows is indicated by the alternations in sedimentary structures associated with changes in flow velocity. Therefore, figure 2E is actually one example of a waxing and waning flow, showing a change from Tc (ripples at its base) to Ta to Tc Bouma intervals. Similar fluctuations in flow velocity can be observed in Fig. 2F, G, H, I and Supplementary Material 2H, L, 3D, E, F, and 4A, B, D.

4. It would be nice to show the bioturbation index to illustrate the sporadic distribution of the trace fossils. Do you see any *navichnia* (mantle and swirl)? Please could you label the key trace fossils and talk about their environmental significance (of course emphasizing the reduced diversity of Cambrian trace makers and ethologies).

We have added a bioturbation index to Fig. 1. We have inserted more information on the trace-fossil content of these deposits for each facies, and we are now including a paragraph summarizing the main ichnologic features as well (lines 298-321; see comment above). A reference to mantle and swirl trace fossils and their significance has been added (lines 207-208, 225-226).

5. White triangles indicating normal grading were added to Fig. 2.

We have incorporated these additions in the figure.

Reviewer #3

This is a well-written, well-argued and informative paper, which presents detailed palaeoenvironmental context for a key fossil site. It should be of broad general interest and is

scientifically sound. I have only a few minor comments (below) and would like to see this published:

We thank the Reviewer for their positive comments.

p. 2, l. 33 – ‘storm floods’ is an ambiguous term, and it is unclear if this means fluvial or marine flooding. The details are explained in the main text but the wording here in the abstract could be more informative.

A brief definition was inserted in the Abstract (lines 33-34)

p. 3, l. 47 – “Stage 5 “should now be Wuliuan.

Corrected

p. 5, l. 108 – it should be nearly impossible to undisputedly identify HCS in a core, lacking the 3D aspect of this structure. This should be reworded to a passive description of the laminae, and emphasise that it is *interpreted* as HCS

The Reviewer is right in that some of the main features of HCS (e.g., wavelength of 1 m) cannot be seen in the core. However, because some of the other diagnostic characteristics (e.g., low angle stratification, laminae mantling the underlying scoured base, internal 2nd-order surfaces) can be seen with confidence in the core, we prefer to keep the term “HCS” at the observational, rather than interpretative level. This is in line with current studies of wave-dominated shallow-marine environments based on structures seen in core material (e.g., Herbers et al., 2016, Canadian Bulletin of Petroleum Geology 64:538—554; La-Croix et al., 2017, Marine and Petroleum Geology 86:736-654; Polo et al., 2018, Journal of Sedimentary Research 88:991-1025) that uses HCS in this sense. Also, it is important to have in mind that HCS has been recognized in coeval strata in outcrop by other authors (e.g., McKenzie et al., 2015) and ourselves.

p. 10, l. 229 – as many tidal signatures can be cryptic, depending on tidal range and asymmetry, it is important to note that absence of evidence is very much not the same as evidence for absence when it comes to tidal sedimentation

We agree with Reviewer 3. We have now slightly rephrased the sentence as “However, no tidally generated structures have been detected in our sedimentologic analysis” (lines 257-258). However, in addition to the core, we have investigated outcrops at different localities, none of which showed a tidal activity. The names of the investigated outcrops are now mentioned in the material and methods section (line 96).

p. 13, l. 299 – could this nearshore area effectively be a nursery for juveniles of open marine species?

In short, yes. The evidence of the dominance of juveniles (alongside high nutrient, and oxygen levels) indicates that Haiyan was a nursery. Yang et al. (2021) suggested this and more specifically that something must have been killing the animals at this site without elaborating much more on the topic. However, what we cannot determine is whether the consistent small size of taxa is attributable to the migration of larger, more mature animals away from Haiyan to avoid the unstable environmental conditions (e.g., such as salinity fluctuations, sedimentation rates, sediment suspension), or perhaps, to animals periodically being killed in the nursery before they reached adult stages. Thus, based on the evidence we have, it is impossible to tell whether animals in Haiyan were in an isolated community completely separated from that of the wider Chengjiang

Biota, or whether Haiyan was the cradle for some (or most) of the Chengjiang community. For this reason, we limit our discussion to the aspects for which we have strong evidence, such as small animal size, fluctuation of sediment flux and so on. In other words, we do not question the nursery interpretation presented in Yang et al., (2021), and we simply shed light on the possible stressful conditions that killed organisms (the existence of these stressful conditions was suggested, but not extensively discussed in Yang et al., 2021).

REVIEWER COMMENTS

Reviewer #1 (Remarks to the Author):

Dear Farid and colleagues,

Thank you for the detailed response to the reviewers and the excellent work on the review. I have no further comments and recommend the manuscript be accepted.

Sincerely,
Julien Kimmig

Reviewer #2 (Remarks to the Author):

Well done..All of my concerns have been fully addressed and I believe the paper is ready for publication as is.

Response to Reviewers:

Reviewer #1 (Remarks to the Author):

Dear Farid and colleagues,

Thank you for the detailed response to the reviewers and the excellent work on the review. I have no further comments and recommend the manuscript be accepted.

Thank you for your review.

Sincerely,
Julien Kimmig

Reviewer #2 (Remarks to the Author):

Well done.. All of my concerns have been fully addressed and I believe the paper is ready for publication as is.

Thank you for your review.